# Effect of Internal Waves on Moving Small Vessels in the Sea

**Andrey Serebryany [1,2]**

1    Shirshov Institute of Oceanology RAS, Moscow 117997, Russia; serebryany@hotmail.com; Tel.: +7-916-1106453
2    Andreyev Acoustics Institute, Moscow 117036, Russia

**Abstract:** Internal waves are responsible for many important processes in the ocean environment (ocean ventilation, energy transfer from large-scale processes to turbulence, etc.). Our goal is to draw attention to the relatively little studied effect of internal waves on ships at sea. We encountered this effect many times while working on the study of internal waves in the shelf zone. The work was carried out from a yacht equipped with the "Rio Grande 600 kHz" ADCP, which makes it possible to measure both the parameters of internal waves and other important parameters of the medium. Two typical examples of the impact of internal waves on a yacht are given. One, when the yacht was at anchor and a train of soliton-like internal waves passed under it. The second, when the yacht was moving and passing over a train of internal waves. Internal waves passing under the moored yacht shifted its position synchronously with the periods of passing waves. A uniformly moving yacht, passing over a package of internal waves synchronously with the period of waves, alternately increased and decreased the speed of its movement. The described effect is explained by the impact on the vessel of the orbital currents of internal waves. Under our conditions, at heights of internal waves reaching 10–15 m, the ship's speed fluctuations reached 0.30 m/s, which was more than 10% of the ship's speed.

**Keywords:** internal waves; moving vessel; ADCP; shelf; sea

## 1. Introduction

Internal waves are a widespread phenomenon in the seas and oceans. It has attracted the attention of researchers for many years. Among the important achievements in studies of internal waves in the ocean, we note the creation of a model spectrum of internal waves by Garrett and Munk [1]. In this work, a large amount of information about the observed frequency spectra of waves was summarized. Further, ocean scientists obtained important data not only one-dimensionally, but also on spatial spectra of internal waves. A great contribution in this area was made by Sabinin and his colleagues, who developed spatial antennas of line temperature sensors and collected data on internal waves in many points of the World Ocean [2,3]. With the introduction of satellite remote sensing techniques into oceanology in the following decades [4–7], it became possible to determine the direction of propagation and other important parameters of internal waves through manifestations on the sea surface. This circumstance has reduced the relevance of measuring spatial spectra of internal waves.

Measurements in the Andaman Sea [8] revealed that intense internal waves exhibit the properties of solitons—nonlinear internal waves capable of propagating over long distances without changing their shape. Measurements carried out in subsequent years have revealed the belonging of intense internal waves to solitons as a widespread phenomenon [9–11]. There are areas of the ocean where internal waves reach large amplitudes and where their impact is most significant. Such areas include water areas near submarine ridges in the ocean [12], as well as large straits [13,14]. Internal waves of large amplitudes are also found in the shelf zone of the ocean in the form of nonlinear tidal waves propagating toward the shore and breaking up into solitons of internal waves. Such areas of the ocean include the northwest shelf of Australia, the shelf of the South China Sea, the Atlantic shelf of the

USA, and other areas [15–19]. Among the many functions for which internal waves are responsible in the ocean, the ventilation of the waters, which is necessary for the existence of life of organisms in the marine environment [20] is of primary importance. In addition, internal waves have a significant influence on sound propagation in the ocean and on underwater acoustics in general [21,22]. A distinctive feature of internal waves is their predominant belonging to the first oscillation mode. Waves of the 2nd mode also occur in the ocean, but such cases are not so frequent [23].

An interesting aspect is also the effect of internal waves on ships floating at sea or structures placed at sea. The surprising fact is that the first report of internal waves in the sea was associated with the first observation of the impact of internal waves on a ship. During the expedition on the Fram in the Kara Sea, Nansen encountered internal waves that created the effect of dead water, which impeded the movement of the vessel [24]. Norwegian oceanographers Bjerknes and Ekman [25] gave an explanation for the phenomenon of dead water, which remains relevant to this day. The essence of the phenomenon lies in the fact that when the ship enters the conditions of a two-layer liquid with a thermocline in the surface layer, it spends energy on the generation of internal waves. This creates a braking effect for the ship. The dead water effect considers the ship as a generator of internal waves, and numerous new results have recently been obtained [26]. However, no one paid attention to the influence of internal waves on floating ships before the start of drilling in the ocean. The vessel conducting deep-sea drilling in the Andaman Sea experienced periodic alternating movements, which made it difficult to carry out the work [27,28]. This revealed effect from deep-sea internal waves forced to carry out special measurements in the drilling area to determine the parameters of the waves and predict their properties [8].

In the shelf zone of the oceans and seas, as mentioned above, intense internal waves in the form of soliton-like wave packets often occur. Horizontal components of orbital currents, accompanying internal waves, reach their maximum values at the boundaries of the marine environment—in the near-surface layer or near the bottom, which can cause a noticeable impact on the objects in this zone. During the SW2006 experiment conducted on the Atlantic shelf of the USA, an unusual phenomenon was noted. In the experiment, a horizontal hydrophone antenna with a length of approximately 400 m was located on the bottom. It was fixed on its ends, but during the passage of a train with intense soliton-like internal waves up to a height of 15–20 m above it, it came in motion, shifting along the bottom by orbital velocities of internal waves [29]. For many years, we have regularly conducted and continued to conduct studies of internal waves in the shelf zones of the seas. For the past 15 years, we have been using small vessels for offshore operations with a set of compact modern oceanographic instruments, among which an important role is played by the Acoustic Doppler Current Profiler (ADCP). This device performs not only the functions of a current meter, but also allows you to measure other important characteristics, such as internal waves, bottom topography, vessel speed relative to the bottom, and etc. The use of ADCP made it possible to identify the effect of internal waves on a small vessel in the shelf zone. In this paper, we consider this effect, which we have encountered many times based on our own experience.

## 2. Materials and Methods

ADCP has become widespread in recent years in oceanographic research. Its usefulness, in particular, for the study of internal waves in the sea, turned out to be obvious. ADCP is a multi-beam pulse sonar with frequency processing of the echo signal. We have carried out numerous measurements of internal waves on the shelves of the Sea of Japan and the Black Sea in the summer–autumn period for many years using a yacht with an ADCP "Rio Grande 600 kHz" installed on it (See Figure 1). The current meter uses the Doppler effect by emitting sound at a frequency of 600 kHz and receiving the echo reflected from sound diffusers in the water. Sound diffusers in the sea are small particles and plankton. These scatterers are almost always present in the marine environment, and their movement in space is carried out by currents. Thus, on average, they move at the same

horizontal speed as water. The frequency of the received echo signal has a Doppler shift due to the flow of water that captured the sound diffusers, which is information about the flow. ADCP measures three components of the current—two horizontal and one vertical relative to a fixed bottom [30]. In addition, it measures the intensity of the backscattered acoustic signal, as well as other important parameters, including the speed and heading of the vessel when moving along the section relative to the bottom.

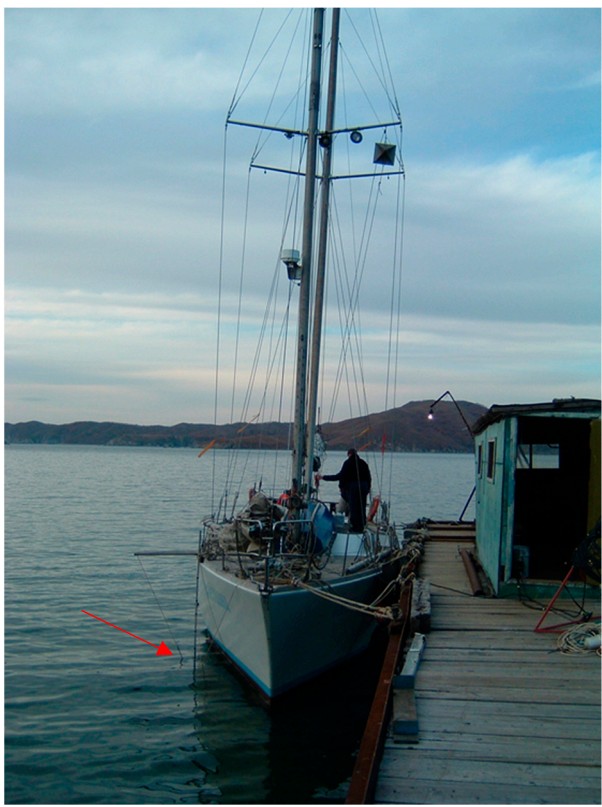

**Figure 1.** A boat with an ADCP installed (its position is shown with a red arrow) for research in the offshore zone.

The ADCP worked in a mode with a cell size of 0.5 m and the interrogation period of only 1 s. When towing with the ADCP, the yacht was moving straight and even with the speed of 2–2.5 m/s. In another measurement, the yacht was anchored at any point on the shelf and in such position that long measurements were carried out with a duration of at least 0.5 or 1 day. The data were recorded on a personal notebook with the WinRiver program installed.

## 3. Results

We consider two examples of an internal wave packet passing under a boat equipped with an ADCP. The first example is when the boat is anchored in the shelf zone, and the second is when the boat is in motion.

### 3.1. Example with a Train of Internal Waves Registered from a Moored Yacht

In the Sea of Japan for several years, we have been conducting studies of currents and internal waves on the shelf of the Peter in the Great Bay. Measurements were made with an ADCP "Rio Grande 600 kHz" installed on the yacht, and two modes were used. This is a survey on spatial sections across the shelf and work on daily stations from the moored yacht. In the Sea of Japan, semidiurnal internal tides were observed, which propagating along the shelf generated packets of intense soliton-like internal waves. These waves passed under the yacht, which made it possible to carry out fairly accurate measurements of their

parameters. Let us consider an example of measuring internal waves from a moored yacht. The yacht was anchored on September 20, 2004 at the shelf point 2 km from the shore where the depth was 40 m.

At about 22:00 local time, a train of five internal waves with heights from 4 to 13 m and periods of 10–19 min passed under the yacht. The largest wave was the third by order of arrival and was 13 m high. The main parameters of the waves are presented in Table 1. Figure 2 shows synchronous records throughout the depth from the surface to the sea bottom at the point of measurements of (a) the echolocation signal (intensity of acoustic backscatter); (b) the vertical component of the current; (c) the magnitude; and (d) the direction of the current. Figure 2e shows a curve of oscillations of the water column current velocity under the boat, caused by passing internal waves.

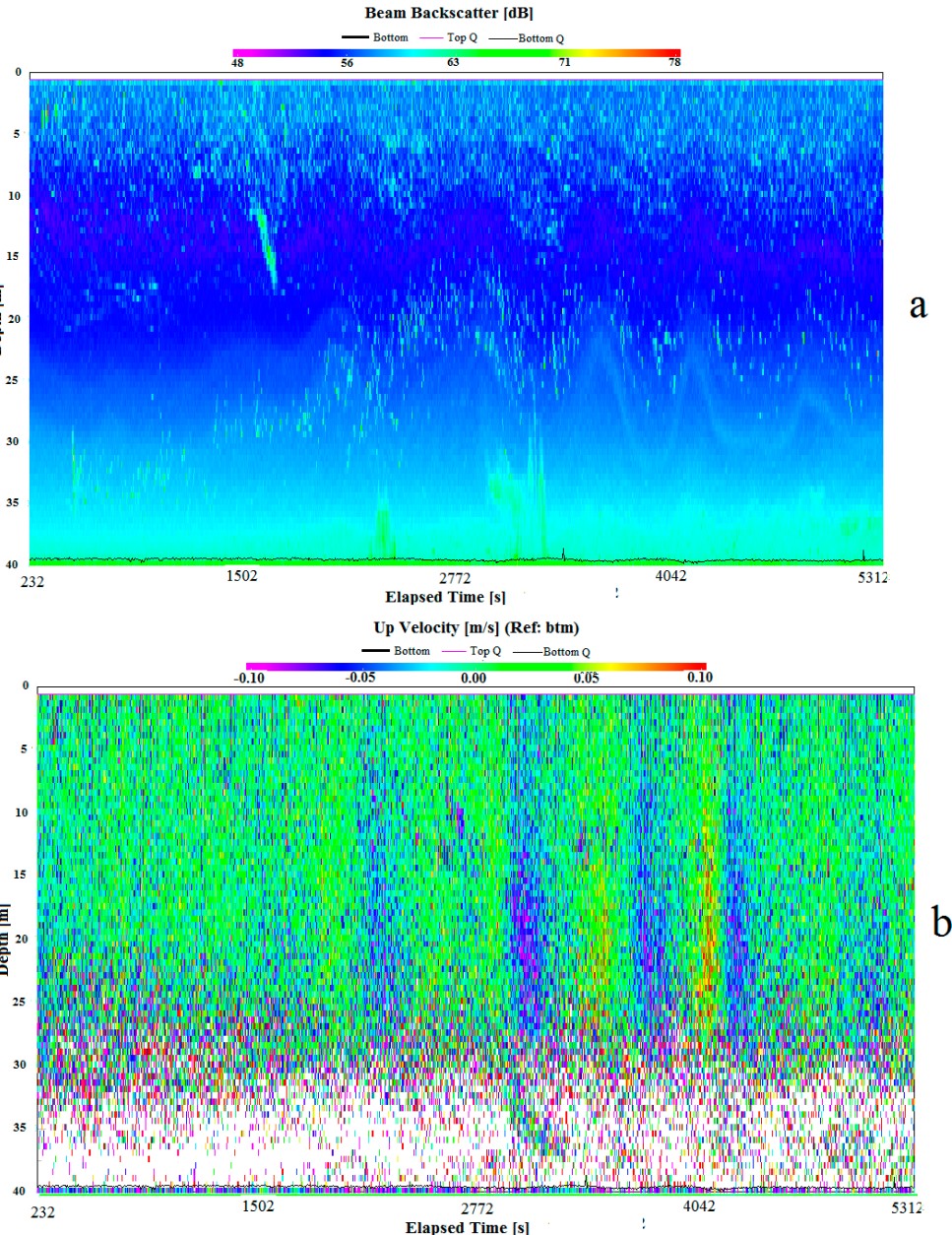

**Figure 2.** *Cont.*

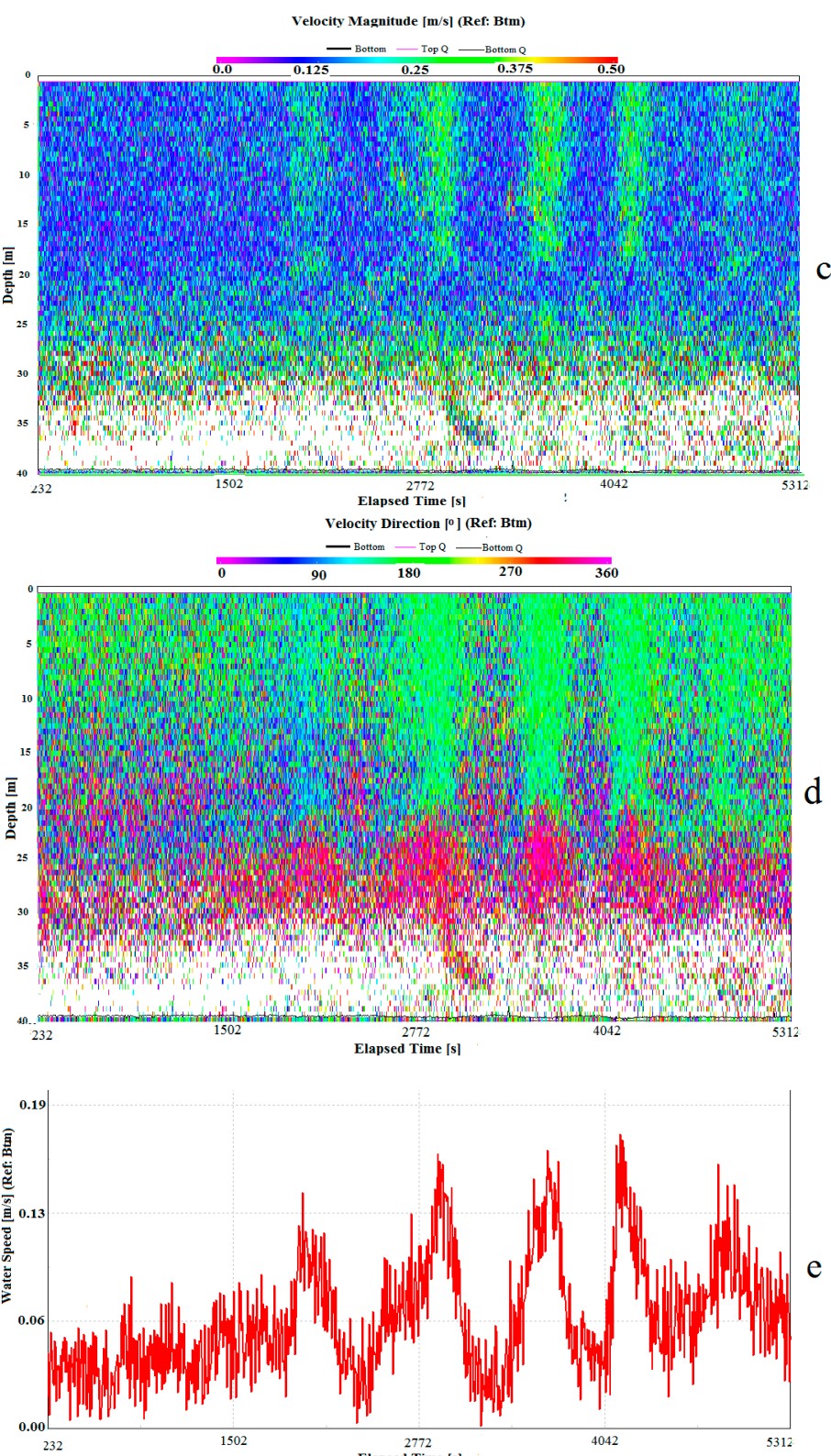

**Figure 2.** ADCP data from a train of internal waves under an anchored boat: (**a**) the echolocation signal (intensity of acoustic backscatter); (**b**) the vertical component of the current; (**c**) the magnitude; (**d**) the direction of the current; and (**e**) the curve of oscillations of the water column current velocity under the boat, caused by passing internal waves.

**Table 1.** Parameters of internal waves and the flow velocity and yacht displacement speed caused by each wave of the train.

| Wave Number in the Train | Wave Height, m | Wave Period, min | Horizon. Flow Velocity, m/s | Yacht Displacement Speed, m/s |
|---|---|---|---|---|
| 1 | 9 | 11 | Crest 0.12 | Crest 0.11 |
| 1 | | | Trough 0.01 | Trough 0.01 |
| 2 | 11 | 14.5 | Crest 0.14 | Crest 0.12 |
| 2 | | | Trough 0.02 | Trough 0.03 |
| 3 | 13 | 12.5 | Crest 0.15 | Crest 0.15 |
| 3 | | | Trough 0.03 | Trough 0.02 |
| 4 | 10 | 19 | Crest 0.17 | Crest 0.16 |
| 4 | | | Trough 0.05 | Trough 0.05 |
| 5 | 4 | 10 | Crest 0.09 | Crest 0.09 |
| 5 | | | Trough 0.05 | Trough 0.02 |

Considering each pattern of water column parameters measured by the ADCP separately, the thermocline was at a depth of about 25–30 m. Its position was well traced by a layer of enhanced backscattering visible on the ADCP backscattering signal. This layer, under the action of passing internal waves, made alternating movements up and down, which was clearly seen on the ADCP record. Thus, the profiles of internal waves were well defined on the time base. In the form of wave profiles, the features of nonlinear waves were clearly visible—sharpened crests and smoothed troughs (See Figure 2a). Internal waves are accompanied by their own orbital currents, which are superimposed on the background current, and the value of which was within 0.05–0.15 m/s. The vertical velocities in the orbital flows of internal waves changed sign from −0.12 m/s (downward motion, blue color palette) to 0.14 m/s (upward motion, yellow-red color palette) (Figure 2b). There is also a periodic increase in the total magnitude of the current at the moments of the passage of internal waves, the largest of which are characterized by values from 0.4 to 0.5 m/s (see Figure 2c). Orbital currents of waves are traced in a change in the direction of the current. In the phase of the passage of the crests, they are directed to the southeast, and when the troughs pass, they change their direction to the opposite, to the northwest (directed towards the coast). The manifestation of orbital flows that change sign once per wave period indicates that internal waves belong to the first mode.

The passage of internal waves, as shown above, causes alternating currents that act on a moored yacht, forcing it to move within distances that are limited by the length of the anchor chain. Figure 3a shows fluctuations in the current velocity during the passage of internal waves, and Figure 3b shows the displacement velocities of the moored yacht recorded using the ADCP. From a qualitative comparison of the data, it can be seen that the displacement velocity of the yacht correlates well with the orbital currents of internal waves that cause it. But for a more rigorous quantitative proof, we will carry out a correlation-regression analysis comparing the data on the speed of the currents and the speed of the displacement of the yacht. The analyzed series included velocity data at the moments of the passage of the crests and bottoms of the train waves. They are presented in Table 1. The analysis performed showed that the relationship between the compared parameters turned out to be very high. The correlation coefficient is 0.980, and the paired linear regression equation is Vdis = −0.301 + 0.952 × Vcur.

Now we will consider what happens to the speed of the yacht when passing over a packet of intense internal waves.

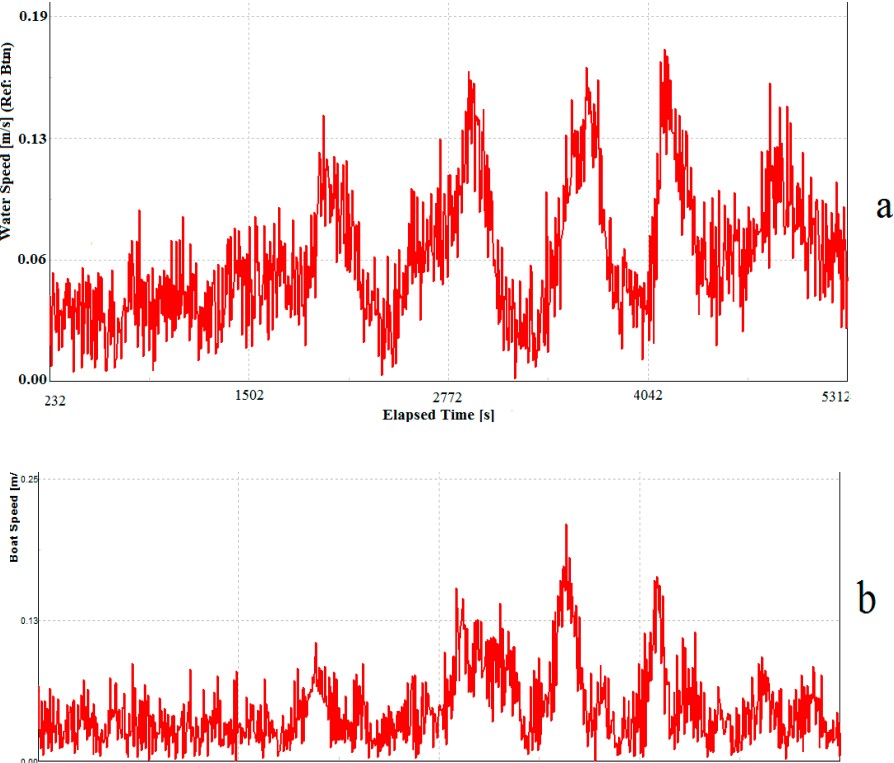

**Figure 3.** ADCP recording of the displacement velocity of the moored vessel (**b**) and the current velocity (**a**) during the passage of the train of internal waves.

### 3.2. An Example of the Passage of a Yacht over a Train of Internal Waves

This example relates to studies carried out on the Black Sea shelf. We then met in the sea a train of record amplitudes for this sea. A train of soliton-like internal waves with heights of 14–16 m was registered, moving from the open sea towards the coast [31]. The reason for the generation of such intense waves in the tidal Black Sea was the passage of a cold atmospheric front over the sea.

The boat was moving along the shelf toward the shore from the open sea in the direction of the tack normal to the shoreline. The motion was equal and straight with a speed of 2 m/s and coincided with the direction of the train of internal waves. The ADCP installed on the boat worked with a sampling rate of 0.8 s (delta time). The vertical cell size was 0.5 m. Figure 4 shows the results of the ADCP recordings. The wave profiles were well recorded in the backscatter data because the orbital currents were intensely turbulent for the inner bottom layers rich in suspended matter (Figure 4a). The vertical components of orbital currents of internal waves reached 0.2 m/s (Figure 4b). The waves moved along the near-bottom thermocline and had the form of internal solitons with sharpened crests and smoothed troughs.

Along with the exceptional parameters of these internal waves, the effect of changes in the boat speed when passing over the wave train is noteworthy. Figure 4c shows the record of the boat speed made by the ADSP synchronously with other parameters of the environment before the encounter with the train and during the passage over it. Figure 4c clearly shows that relatively stable ship's speed before encountering the train and when it begins to experience modulation with periodic acceleration and deceleration. The spatial period of these changes coincides with the horizontal size of the internal waves, and the range of vessel speed fluctuations reaches 0.3 m/s.

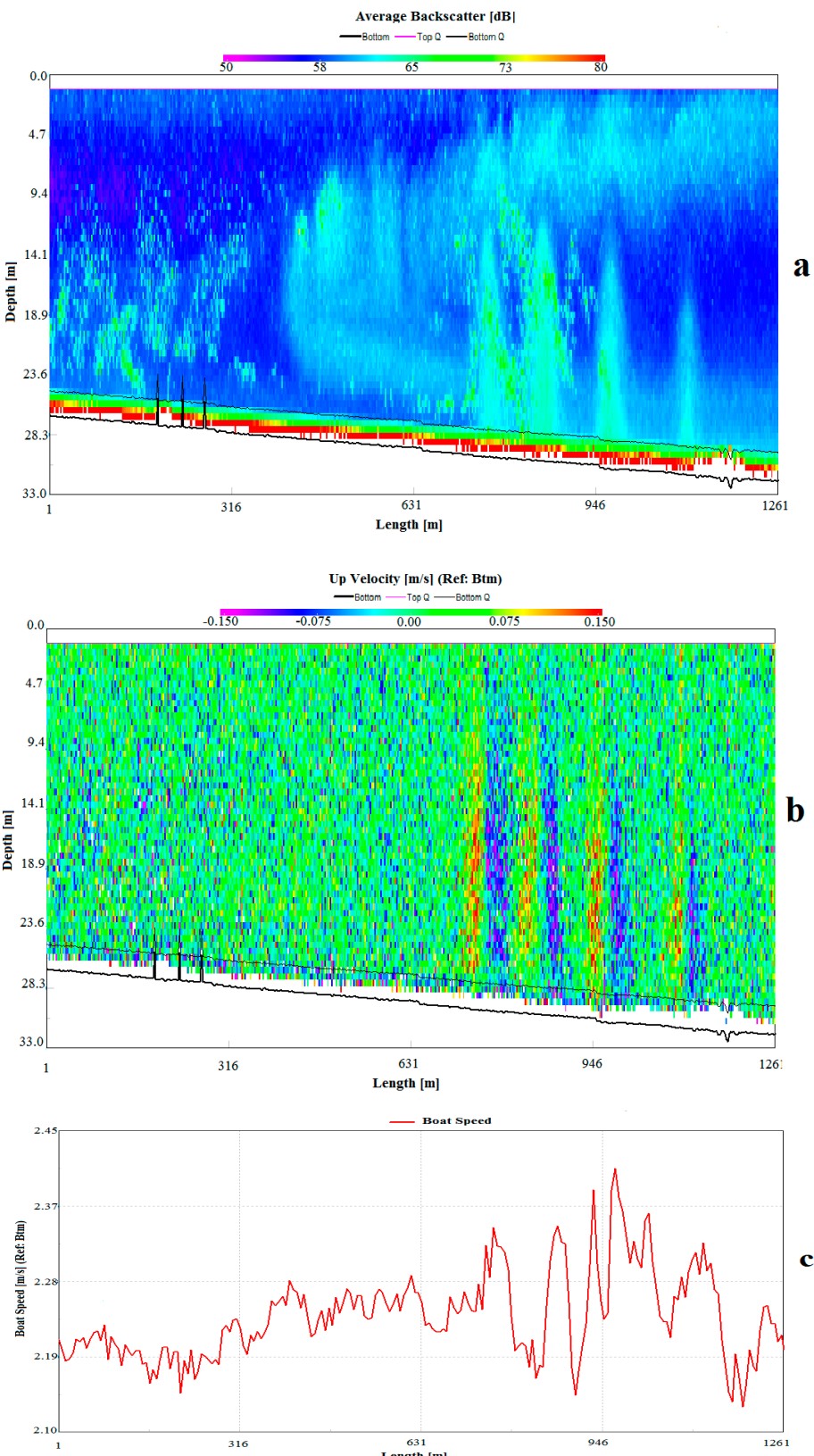

**Figure 4.** ADCP recordings on a rectilinear spatial section when passing over a train of internal waves: (**a**) picture of echolocation contrasts (acoustic backscatter signal); (**b**) data of vertical current velocity; and (**c**) data of the vessel motion speed relative to the bottom.

## 4. Discussion

We have presented observational data that testify to the influence of internal waves passing under ships of small tonnage in the shelf zone of the sea. These waves with typical amplitudes of 10–15 m have a noticeable effect on the speed of the ship, and can also adversely affect the course of the ship. The measurements were carried out in the summer, when the seasonal thermocline was at depths close to the shelf bottom. Accordingly, the displacement maxima in internal waves were located at the near-bottom area. However, the orbital currents of internal waves in the near-surface layer of the sea, despite this, were large enough to affect the ship. The essence of the effect is simple. It lies in the fact that a floating or drifting ship falls into the area of influence of internal waves passing under it due to the fact that the orbital currents of these waves go into the near-surface layer of the sea and alternately create deceleration and an increase in the speed of the ship. But in the ocean there are places where there are internal waves of large amplitudes, reaching several tens of meters and even hundreds of meters. In these cases, the effect of internal waves on ships in their sphere of influence becomes more significant. Therefore, on the Mascarene Ridge in the Indian Ocean, where internal waves with a height of almost 100 m were encountered, the effect of modulation of the ship's speed during the passage of internal waves was also observed [32,33]. It should be noted that it was a large ship; it weighed 10 tons. The effect of ship speed modulation is connected to the action of orbital currents accompanying the internal wave. The greater the magnitude of the orbital currents of internal waves, the greater the effect. When the vessel moves along a course coinciding with the direction of propagation of the internal wave or a course opposite to the wave, the effect is most pronounced. There is a periodic increase and deceleration of the ship's speed.

The influence of large internal waves of the same region on the hydrodynamic deepener during towing using a line temperature sensor was also studied [34]. It was revealed that due to significant orbital currents accompanying intense internal waves, the deepener makes vertical movements during towing "in phase" with the internal waves it crosses. The vertical displacements of the deepener were within 3–11 m at the intersection of internal waves with heights of 10–80 m. This effect took place both in the case of the intersection of solitary internal waves and packets of short-period internal waves. The considered effect is important for any towed and independently moving underwater object. At present, when uninhabited and manned underwater vehicles are actively involved in the practice of oceanic and marine research, it is necessary to take into account the difficulties that may arise when they encounter high-amplitude internal waves.

## 5. Conclusions

Thus, we considered examples of measurements of internal waves on the sea shelf using ADCP in which the effect of internal waves on the speed of small tonnage ships was revealed. This effect is due to the influence of orbital currents, which always accompany internal waves. It leads to a noticeable modulation of the vessel's speed. The modulation of the ship's speed is in phase with the parameters of the internal waves, resulting in a periodic increase and slowdown in the ship's speed. The effect manifests itself to the maximum when the course of the ship coincides with the direction of propagation of internal waves. Under our conditions, at heights of internal waves reaching 10–15 m, the ship's speed fluctuations reached 0.30 m/s, which was more than 10% of the ship's speed. To achieve the goal of accurately measuring internal wave parameters, it is necessary to take into account the possibility of encountering this effect if you are working from a towing or drifting vessel. This effect is most significant for large amplitude internal waves.

**Funding:** This research was funded by a State Assignment of the Ministry of Science of the Russian Federation (project no. FMWE-2021-0010) and by the Russian Foundation for Basic Research (project nos. 19–05–00715).

**Data Availability Statement:** The data presented in this study are archived at the Ocean Acoustics Laboratory of Shirshov Institute of Oceanology, Russian Academy of Sciences and available on request.

**Acknowledgments:** TRD Instruments-Europe provided author with a Workhorse Rio-Grande-600 kHz acoustic Doppler current profiler for temporary use.

**Conflicts of Interest:** The author declares no conflict of interest.

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
