# Peer review of "Effect of Internal Waves on Moving Small Vessels in the Sea"

_fluids, doi:10.3390/fluids8020075_

Round 1

Reviewer 1 Report

The author analyzed two records obtained by ADCP in the shelf zones of the Japan and Black Seas to show the effect of internal waves on small vessels.  In short, the problem considered is interesting and the manuscript is clearly written. I am in favor of its publication in Fluids if the following issues could be addressed.

1 In the abstract, the author is suggested to provide the motivationthe main results and conclusions of this work in a briefly manner. No background introduction is needed in the abstract.

2 It is too hard to see clearly the axis label in figure 2 to 4. Please enlarge the font size of the axis label.

3 The author shows qualitative analysis of the obtained data. Quantitative analysis is suggested to consider in this work to help draw the conclusion.

Reviewer 2 Report

The author analyzed a number of records obtained during the passage of small vessels  over intense internal waves in the shelf zones of Japan and Black Seas. The objective of the work is interesting; however, the manuscript is not well-written, and the presented results are not of scientific value and of interest to other researchers. My comments are as follows:

·   The abstract and Introduction should be written clearly by focusing on the novelty of the present study. 

·         Details of the experiments should be presented briefly.

·         The discussion about the results should be improved.

·         More results are needed to achieve the objective of the paper.

·         There are many grammatical errors.

Reviewer 3 Report

  1. Please make your literature review more rigid, and include some recent references on the topic
  2. Results should be numbered Sect. 3 (line 88)
  3. Expand ADCP - explain what it is.
  4. Nothing is visible in Fig. 2 - the numbers or the text. The figure should be redone for clarity. Simply captioning the figure as “ADCP data” does not help. What are the subplots a-e? I cannot read the text in this figure even after magnifying the pdf.
  5. Line 108: “Around 22 h a packet of five internal waves with heights from 13 to 4 m and periods of 19-10 minutes passed under the yacht.” - this sentence is not at all clear. What is 22 h? Why do you specify the heights and periods in descending order?
  6. How do you measure the horizontal flow velocity in table 1? What location/time does it correspond to?
  7. I recommend using some other terminology for wave number in table 1 - wavenumber (with no space) has a clear meaning, so it is better to use something else, albeit you have a space between wave and number
  8. Since figure 2 is of poor quality, I am not able to follow the text in lines 120-132
  9. The text and numbers in figure 3 are also hard to read. Please change all your figures and make them sharp and readable 
  10. What is the basis on which you claim in line 142 that “the velocity of the yacht's displacement correlates well with the orbital currents of internal waves that cause it”? This must be clearly explained.
  11. The discussion section is very weak - in fact, there is hardly any discussion! This section continues (from the pervious one) to list some observations, and highlight some trends in variation of some parameters. The flow physics is not discussed at all - I think it is crucial for all papers submitted to Fluids to discuss important physics.
  12. The novelty of this work is not clear at all. What is the contribution of this work to understanding the effects of internal waves on moving vessels? What are the major applications of this work?

The paper does not help in understanding the underlying physics of this problem, and the discussion is superficial. The figures are of poor quality, and this makes it a challenge to read and understand the paper. 

Round 2

Reviewer 2 Report

The author has answered and incorporated all the suggestions satisfactorily. Hence the publication is recommended.